# Casting Simulation of Large-Volume Fluid Cementitious Materials: Effect of Material Properties and Casting Parameters

**DOI:** 10.3390/ma16216856

**Published:** 2023-10-25

**Authors:** Junkai Wang, Shenyou Song, Wen Xu, Lizhi Zhang, Guodong Xu

**Affiliations:** 1Jiangsu Key Laboratory of Construction Materials, School of Materials Science and Engineering, Southeast University, Nanjing 211189, China; 220202231@seu.edu.cn (J.W.); xuwen@cnjsjk.cn (W.X.); 230228658@seu.edu.cn (L.Z.); 2Shenzhen-Zhongshan Link Administration Center, Zhongshan 528400, China; 3State Key Laboratory of High Performance Civil Engineering Materials, Nanjing 210008, China; 4Jiangsu Research Institute of Building Science Co., Ltd., Nanjing 210008, China

**Keywords:** self-compacting concrete, casting, simulation, compactness, segregation

## Abstract

The increasing pressure of traffic congestion on socio-economic development has made the construction of cross-water transportation ever more crucial. The immersed tunnel method is among the most extensively employed. However, a critical challenge of the immersed tunnel technique is to ensure the compactness and stability of concrete during the casting process. Conventional laboratory methods face challenges in achieving large-volume concrete casting, resulting in the notable waste of human and material resources. Hence, this study employs a simulation approach to investigate the casting parameters and the fresh properties of concrete, exploring their impacts on concrete stability and compactness. The results indicate that when the surface tension of concrete exceeds 0.03 N/m, and the yield stress and plastic viscosity are 50 Pa and 50 Pa·s, respectively, the concrete exhibits excellent casting compactness. A design incorporating three large and six small outlets, paired with a casting speed of 3 cm/s, achieves superior compactness. Additionally, when the yield stress of concrete exceeds 3 Pa, there is no segregation of aggregates. In cases where segregation occurs, the thixotropic property of the cement paste contributes to a significant reduction in the velocity of aggregate segregation.

## 1. Introduction

With the rapid development of the socio-economy, there is an increasing demand for navigation both on and under water. The traditional method is to build bridges to facilitate the flow of people and goods, which is only suitable for low-demand waterways [1]. In addition, bridges have the disadvantage of occupying large spaces and requiring large areas, so there is a pressing need for a new way of navigation. Meanwhile, due to the pressure of traffic congestion also increasing, the traditional way of building bridges has been difficult to solve, so the establishment of cross-waterway transportation is particularly urgent [2].

Underwater tunnels have become a preferred choice for crossing rivers and seas due to their all-weather capabilities, large capacity, convenience, and low impact on shipping and the environment, among other navigation options [3,4]. Common construction methods for underwater tunnels include drilling and blasting, shield tunneling, the caisson method, the water-separation open-cut method, and the immersed tube method [5]. Among them, the immersed tube method involves prefabricating the tube sections and completing the connections underwater. This is adaptable to various geological and hydrological conditions with a large carrying capacity, and is cost-effective, making it the first choice in economically developed countries and regions [6].

In 1910, the Central Railway Tunnel in the Detroit River in the United States was the first steel shell immersed tube tunnel prepared using the immersed tube method [7]. Over the next 30 years, the U.S. constructed most of the world’s immersed tube tunnels, primarily using steel-shell concrete structures. And many tunnels were built in the U.S. bays, where the water depth is generally deeper than inland rivers. Considering the force on the immersed tube, the immersed tube often uses circular or elliptical steel shells [8]. During this period, the main load-bearing structure of the immersed tube tunnel was concrete, with thin steel shells used for waterproofing and serving as molds during concrete casting, without an effective shear connection between the steel and concrete. 

Starting in the 1960s, Japan introduced a steel–concrete–steel composite structure [9]. In this structure, the internal concrete prevents the steel plate from buckling, while the external steel shell prevents the concrete from cracking. Moreover, the tunnel consists of a number of cavities with typical dimensions of 3.5 m × 3 m × 1.5 m by length, width, and height. This structure enhances the load-bearing capacity of the tunnel and gives it better ductility, impact resistance, waterproofing, and other qualities [10]. Moreover, to accommodate road traffic development, most Japanese immersed tube tunnels have adopted rectangular cross-sectional structures with high utilization rates. 

Drawing from Japan’s technology, China, in similar circumstances, constructed the Hong Kong–Zhuhai–Macau Bridge Undersea Tunnel and the Shenzhen–Zhongshan Undersea Tunnel [11,12]. The latter, for example, overcame five major technical challenges: extreme width, variable width, deep burial, large silt backfill, and poor stability in the sand mining pit area. This achievement enabled the construction of a 6.8 km long tunnel, greatly facilitating traffic and logistics transportation.

The successful application of these steel shell-immersed tube tunnels is closely related to the development of self-compacting concrete (SCC) [13,14]. Ordinary concrete requires vibration to achieve the desired compaction, while SCC has excellent flowability and can fully compact into every corner of the mold through its weight, laying the foundation for the construction of steel shell concrete immersed tubes. However, factors like viscosity, yield stress, and aggregate properties can affect its flowability, segregation, and air content [15,16,17]. Additionally, the layout of air outlets during casting is also a critical factor in determining the bonding between the steel plate and concrete, thus affecting the working performances of a steel shell-immersed tube tunnel [18]. 

Many researchers have launched a series of experimental studies on the flowability of SCC and the casting process of steel shell-immersed tube tunnels. For example, Wang [19] studied the working performance of SCC with different mineral admixtures and examined the influence of concrete casting speed and the number of outlets on the compactness of the chamber; Rodrigo Andraus [20] studied the effects of admixtures on the flowability of SCC; and R. Zerbino [21] investigated the effects of external factors (like concrete temperature and environmental exposure conditions) on the flow properties of SCC and determined the yield stress and plastic viscosity of different types of SCC. However, casting these large volumes of concrete is difficult to complete during the experimental conditions and is a significant waste of manpower and resources. Compared to the highly restrictive laboratory testing environment, numerical simulation can yield more accurate results based on a rigorous theoretical background. At the same time, numerical simulation can further simulate extreme conditions that experiments cannot achieve and can provide reasonable guidance for experimental testing. Therefore, using simulation means to simulate the concrete casting process is of great significance.

With the popularization of theoretical models of concrete and computational fluid dynamics (CFD) analysis, numerical simulations of concrete flow have become more mature. Currently, numerical simulation methods for the simulation of concrete flow mainly include Fluent, STAR-CCM+, CATIA, and so on. Based on this, Thrane [22] used Fidap software (version 8.0) to conduct two-dimensional and three-dimensional numerical simulations of the flow of SCC in an L-type instrument test; Frederic Dufour [23] used the finite element method with Lagrangian integration points and the Bingham model to analyze the flowability of fresh concrete over time and space. Although these software programs have faster operation rates for single-field calculations, the concrete is a complex mixture, requiring considerations like two-phase flow between concrete and air and the relative motion between the interstitial cement paste and aggregates in concrete mixtures. Therefore, multi-physics field simulation is involved. COMSOL software (version 6.1) has a significant advantage in multi-physics field simulation, accurately handling couplings between multiple physical fields and offering a more comprehensive custom interface for fluid parameters and grid division. In addition, each operation module has its own built-in formulas, which greatly simplifies the modeling process.

Therefore, this paper uses COMSOL finite element software (version 6.1), with secondary development, to study the influence of material properties and casting parameters on casting compactness within SCC mixtures and between steel plates and SCC, as well as the aggregate segregation of SCC. The parameters involved in this paper are the yield stress, plastic viscosity, surface tension of SCC, casting speed, number, and position of air outlets, etc.

## 2. Theoretical Background of Concrete Casting Simulation

Since E.C. Bingham and others first proposed the Bingham fluid, it has been applied to materials such as porcelain clay and diatomaceous earth, characterizing the material properties through the definition of yield stress and plastic viscosity. Subsequent research by Tattersall and Banfill [24,25] found that the Bingham model can accurately describe the rheological performance of fresh mixed SCC. In addition, the main models currently describing the rheology of concrete include the Herschel–Bulkley model [26], Casson model, and so on. Among them, the Bingham model is widely used, which only needs to define two parameters and can relatively accurately describe the flow of concrete. 

However, according to the rheological curve of the Bingham model, as the strain rate γ approaches zero, the slope of the curve which represents the plastic viscosity μ tends toward infinity. This phenomenon poses challenges for numerical computations. Therefore, Papanastasiou [27] proposed a regularized Bingham model. This model introduces some correction factors based on the Bingham model, which can more accurately describe the flow of concrete, and by changing the parameters, it can make the rheological characteristics of fluid conform to Newtonian or non-Newtonian fluids. In this paper, based on the COMSOL finite element software (version 6.1), the Bingham–Papanastasiou model was used to describe the flow of SCC. The model equation is as follows:(1)$$\tau =\tau_{0}\left(1-e^{-m\gamma }\right)+\mu$$
where $\tau_{0}$ is yield stress of concrete, m is a model parameter (according to the research by Jeong [28] and others, it was found that when m = 100, it satisfies the Bingham fluid). Therefore, in this paper, the model parameter *m* is set to 100. 

The casting mold is shown in Figure 1. To simplify the mold and increase the simulation calculation efficiency, a small-scale mold (with length, width, and height scaled down to 1/10 of the full-scale mold) was used to replace the full-scale mold, and only a two-dimensional mold was simulated. In addition, since the mold is axisymmetric, only half of the mold was used for the analysis in subsequent simulations. 

The “Two-phase flow-Laminar” model was employed to treat the fresh mixed SCC as a homogeneous Bingham fluid, examining the effects of the number of air outlets, concrete surface tension, rheological properties, and casting speed on casting compactness. In this model, the phase field method is used to capture the air–water interface when solving the Navier–Stokes equations. In addition, the interface position is tracked by the phase field variable equation and the mixing energy density equation [29]. The model was meshed with a free tetrahedral mesh with 33,792 grid cells. A segregated solver was used to solve the velocity field, pressure field and phase field variations separately. The computational time step was 0.05s, and the solution error was 10^−2^, which had good convergence. 

The density of the concrete was set to 2400 kg/m^3^. A gravel size of 20 mm was chosen, with a density of 2800 kg/m^3^. The water-to-cement ratio of cement paste was set to 0.35. The contact angle between the concrete and the steel plate was set to 45°.

The yield stress and plastic viscosity of SCC were set to be 50 Pa and 50 Pa·s as a reference, according to Wallevik’s study [30]. Wallevik, in his study, plotted areas showing different types of SCC. A yield stress of 50 Pa and plastic viscosity of 50 Pa·s are within the recommended range, based on which the values of plastic viscosity and yield stress of 50 Pa·s and 50 Pa are suitable for practical engineering.

Additionally, aggregate movement in concrete affects the performance of concrete, as well as its construction stability and safety. In recent years, in the numerical simulation studies of liquid–solid two-phase flow, the Euler–Euler method and the Euler–Lagrange method are commonly used [31]. Among them, the Euler–Euler method treats the particles and fluid as interpenetrating and continuous fluids to deal with the calculation. Similar to the form of single-phase flow equations, which makes the form of the particle-phase equation set consistent with the form of the fluid-phase equation set, the Euler–Lagrange method, therefore, can be used in a unified form and solution method. However, due to the assumption of the existence of the proposed fluid in the two-fluid model, it is difficult to obtain the trajectory of the particle phase, and the Euler–Lagrange method makes up for its shortcomings [32].

The Euler–Lagrange method treats the fluid phase as a continuous phase and the particle phase as a discrete system, which allows the trajectories of all the particle phases to be tracked. On the one hand, the Eulerian method is used to deal with the motion of the fluid phase by borrowing the form of the single-phase flow equations to calculate the equations of control of the motion of the fluid, and using the Navier–Stokes equations of the coupled liquid-solid two-phase system to describe the laws of motion of the fluid phase [33]. On the other hand, the Lagrange method is used to calculate the trajectories of individual particles, so the motion law of the whole particle phase can be analyzed by counting the trajectories of a large number of particles. Therefore, in this paper, the “Euler-Lagrange” model is utilized to study the movement of aggregates within the concrete. The equations are as follows:(2)$$\frac{\partial }{\partial t}\left(\varepsilon_{f}\rho_{f}\right)+\nabla \left(\varepsilon_{f}\rho_{f}u_{f}\right)=0$$
(3)$$\frac{\partial }{\partial t}\left(\varepsilon_{f}\rho_{f}u_{f}\right)+\nabla \left(\varepsilon_{f}\rho_{f}u_{f}u_{f}\right)=-\varepsilon_{f}\nabla p-\varepsilon_{f}\nabla \tau_{f}+\varepsilon_{f}\rho_{f}g-F_{int}$$
(4)$$m_{p}\frac{dv_{p}}{dt}=\sum F$$
where $\varepsilon_{f}$ is the liquid-phase volume fraction, $\rho_{f}$ is the liquid-phase density, $u_{f}$ is the liquid-phase velocity, *t* is the time, p is the liquid-phase pressure, $\tau_{f}$ is the liquid-phase stress tensor, $F_{int}$ is the liquid–solid interphase force, $m_{p}$ is the mass of the solid particle, and F is the total force on the particles.

As long as steady-state flow is reached, the behavior of fresh concrete can be described using a yield stress model such as the Bingham or Hershel Bulkley models. However, in the case of cementitious materials, things are not so simple as the hydration process starts as soon as cement and water are mixed together. The apparent viscosity of the material is permanently evolving, as described by Otsubo et al. [34]. This phenomenon leads to the thixotropic feature of concrete, where the mixture seems to flocculate rather quickly at rest and it becomes apparently more and more fluid while flowing during typically several tens of seconds. Since all concretes are thixotropic, in this paper, the thixotropic model proposed by Roussel [35] is used to more accurately describe the movement of aggregates in the cement paste, and the model equations are as follows:(5)$$\tau =\left(1+\lambda \right)\tau_{0}+\mu \gamma$$
(6)$$\frac{\partial \lambda }{\partial t}=\frac{1}{T\lambda^{m}}-\alpha \lambda \gamma$$
where μ is plastic viscosity, λ is the flocculation state of the material, and *T*, *m* and *α* are thixotropic parameters.

## 3. Simulation Results of Small-Scale Mold Casting

### 3.1. Effect of Outlet Number and Casting Speed

In this study, a small-scale mold was first used to examine the general impact of the number and layout of outlets on casting compactness. The effect of one outlet on casting compactness was first studied, as shown in Figure 2a, where the distance from the outlet to the casting inlet changed. It was observed that when the casting inlet was far from the outlet, the casting compactness was acceptable, with no evident debonding area, which refers to the debonding of the concrete and steel shell wall due to the presence of air. However, when the distance ratio (*d*/*h*, where *d* is the distance between outlet and mold boundary, and *h* is the distance between the inlet and mold boundary) exceeded 0.2, a noticeable debonding zone appeared in the upper-left corner of the mold, and the closer the outlet was to the casting inlet, the larger the debonding zone. 

Based on this, an additional outlet was introduced to study casting compactness, as shown in Figure 2b. Keeping the outlet on the left constant, the right outlet was progressively moved away. It is found that when the two outlets were very close, the casting compactness was excellent, with no obvious debonding zones. However, when *d*/*h* exceeded 0.4, a noticeable debonding area appeared between the two outlets, and the debonding area increased as *d* increased. Consequently, three outlets were used, as shown in Figure 2c. It was observed that regardless of how the distance between the three outlets changes, the casting compactness remained relatively good, with no evident debonding area.

Next, using the three outlets as an example, the influence of casting speed on casting compactness is studied, as illustrated in Figure 2d. Speeds from 5 mm/s to 20 mm/s are examined. It is found that when the speed is relatively low (below 10 mm/s), the casting compactness is excellent. However, as the speed gradually increased, an evident debonding area appeared in the final mold. This is mainly because, at higher speeds, the SCC is more prone to entraining air bubbles, and the SCC does not have sufficient time to flow uniformly, resulting in a debonding area.

### 3.2. Effect of Yield Stress and Plastic Viscosity

SCC can be considered as a Bingham fluid with yield stress and plastic viscosity ranging from 25 Pa, 25 Pa·s to 100 Pa, 100 Pa·s, respectively [30,36,37]. The effects of the yield stress and plastic viscosity on the casting compactness and air content were investigated. As shown in Figure 3a, the plastic viscosity of SCC was 50 Pa·s, and yield stress changed from 25 to 100 Pa. When the yield stress was 25 Pa and 50 Pa, there was no debonding area between SCC and the mold. However, when the yield stress increased to 100 Pa, it was obvious that there was a debonding area between SCC and the mold. This was mainly because it is hard for SCC to flow as the yield stress increases, so it is easy to form a debonding area between SCC and the mold. 

Moreover, the effect of plastic viscosity on casting compactness was investigated, as shown in Figure 3b. The yield stress was 50 Pa, and the plastic viscosity changed from 25 to 100 Pa·s. It can be seen that there was no apparent difference in casting compactness as the plastic viscosity changes, with no debonding area between SCC and mold at each viscosity. It can also be seen from Figure 3b that the changes in plastic viscosity can only affect air content. When plastic viscosity increased from 25 Pa·s to 100 Pa·s, the average air content changed from 0.34 to 0.31. This is mainly because the higher plastic viscosity was more difficult to entrap in air, leading to a relatively low air content. However, when the yield stress and plastic viscosity were both low, it was easier to introduce air, which is not favorable for practical engineering. Therefore, a medium value of plastic viscosity and yield stress, i.e., 50 Pa·s and 50 Pa, is suitable for casting.

### 3.3. Influence of Surface Tension

The air content of SCC under different surface tensions was investigated, and four different surface tensions of 0.0001 N/m (which is an extreme case), 0.02 N/m, 0.03 N/m, and 0.063 N/m were simulated; the results are shown in Figure 4. Firstly, as shown in Figure 4a, when the surface tension approached 0, the SCC easily produced air and there was high air content. This was mainly because the high viscosity of the SCC and the excessive air content made it impossible to eliminate air. Therefore, air aggregation occurred, leading to the creation of debonding areas between the mold and SCC. When the surface tension was 0.02 N/m, as shown in Figure 4b, a similar phenomenon could be seen. In this case, a similar phenomenon of the debonding zone occurred, which was due to the surface tension being too low; the air from the surface overflows and, thus, aggregates in the inner top of the mold, resulting in the generation of a debonding area. 

When the surface tension was further increased to 0.03 N/m and 0.063 N/m, as shown in Figure 4c,d, the SCC is well compacted, and there is no obvious debonding zone, but the air content of the SCC at 0.063 N/m surface tension was about 3%, which was slightly higher than that at 0.03 N/m (about 2.5%). This is mainly because the appropriate reduction in the surface tension of the system can lead to the formation of a new interface with low surface energy, thus reducing the strength and stability of the bubble film, and accelerating the exclusion of the liquid around the bubbles [38,39]. Therefore, when the surface tension is higher than 0.03 N/m, it exhibits excellent casting compactness, while an appropriate reduction in the surface tension of concrete is suggested to enhance casting compactness.

## 4. Simulation Results of Full-Scale Mold Casting

### 4.1. Effect of Number of Outlets and Casting Speed

The difference between the full-scale mold and the small-scale mold was further investigated based on the small-scale mold. Since the full-scale mold is large and requires a lot of computational resources, we first chose the full-size with a high casting speed (20 cm/s), as shown in Figure 5. It can be found that the size of the largest air void was about 10 cm. It should be noted that it is the debonding zone between SCC and steel plates that is of concern, rather than the air void within the bulk-poured SCC. Hence, we mainly focused on the content at the top of the mold. Therefore, considering the largest air void, the mold was simplified and the height of the mold was set to 20 cm to save computational resources.

The effect of the number of outlets and their layout on the casting compactness was explored, as shown in Figure 6. Keeping the casting speed constant at 3 cm/s, it can be seen that the air content decreases with the increase in the number of air outlets, and the ratio of the debonding zone was about 13% of the molded casting volume when there was only one air outlet. When there were two outlets, the ratio of the debonding zone was about 6%. When there were three outlets, the ratio reduced to 2%. It can be found that although the debonding zone decreased as the number of outlets increases, there was still a small amount of the debonding area. 

Moreover, it is obvious that there is a difference between the results of the simplified full-scale mold and those of the small-scale mold, as shown in Section 3.1, mainly because the spacing of the outlets in the small-scale mold is still very small compared with the full-scale mold, even though the layout of the outlets is also arranged according to them in small-scale mold. Therefore, the full-scale mold is not equivalent to the small-scale mold.

Moreover, the discussion on the effect of the casting rate on content was further analyzed. Two different velocities were chosen, 3 cm/s and 0.75 cm/s, as shown in Figure 7a,b. It can be found that when the casting velocity was reduced, there was still a debonding zone. Therefore, we continued to add small outlets on this basis to increase the air release capacity of outlets. As shown in Figure 7c, it can be found that the SCC can fill the whole mold after adding six outlets, which meets the actual engineering requirements.

### 4.2. Influence of Yield Stress of Cement Paste on Aggregate Stability

According to N. Roussel’s [40] study on the interaction between particles and Bingham fluid, the segregation tendency of particles depends on the yield stress of the Bingham fluid. Meanwhile, the segregating velocity of particles has a relationship with the plastic viscosity of Bingham fluid [41,42]. When the solid fraction and the yield stress of the Bingham fluid satisfy the relation shown in Equation (7), the particles can be regarded as stable in Bingham fluid without segregation.
(7)$$\emptyset \geq \emptyset^{*}{\left(\frac{9\tau_{0}}{2\left|\rho_{s}-\rho_{f}\right|dg}+1\right)}^{-3}\;$$
where $\emptyset^{*}$ is a maximum solid fraction ($\emptyset^{*}=0.8$), $\rho_{s}$ and $\rho_{f}$ are densities of particle and cement paste, respectively ($\rho_{s}=2800\;kg/m^{3},\rho_{f}=2100\;kg/m^{3}$ in this work), *d* is the size of the particle (*d* = 2 cm in this work),and *g* is the gravitational acceleration (*g* = 9.8).

The expected value of $\emptyset /\emptyset^{*}$ is no less than 0.75 for a well-flowing SCC. Then, when $\emptyset /\emptyset^{*}=0.75$, $\tau_{0}$ can be calculated from the above formulation ($\tau_{0}=3$ Pa). Therefore, it can be known that when the yield stress is grated than 3 Pa, aggregates can be stable in the cement paste. In contrast, when the yield stress is less than 3 Pa, aggregates will segregate in the SCC. 

Therefore, the yield stresses of 2 Pa and 4 Pa are selected to study the segregation performance of aggregates. It can be found that the aggregates do not have any segregation and are stably suspended in the cement paste when the yield stress of the cement paste is 4 Pa, as shown in Figure 8a. As shown in Figure 8b, when the yield stress is 2 Pa, it can be found that the aggregate is segregated. The main reason is that when the trailing force exerted by the cement paste on the aggregate is insufficient, the driving force exceeds the trailing force, then the aggregate will start to segregate. Therefore, it can be concluded that when the yield stress of the cement paste is more than 3 Pa, the largest aggregate can be stabilized in it, so the smaller-diameter aggregate can also be stabilized in the cement paste.

### 4.3. Influence of Thixotropic Property on the Segregating of Aggregates

The thixotropic and Bingham models were used simultaneously to describe the effect of changes in cement paste viscosity on aggregate segregation. The simulation results are shown in Figure 9. The viscosity of cement paste was 2 Pa·s and 8 Pa·s in Bingham models, and it can be found from Figure 9 that with the increase in paste viscosity, the segregating distance of particles gradually decreased. This was mainly because the high-viscosity paste has more resistance to the particles, so the segregating speed will be slower. 

In addition, it can be seen from Figure 9 that the results of the thixotropic property demonstrated a great difference from the results without the thixotropic feature of the cement paste (the paste viscosity was 2 Pa·s when the thixotropic property was considered), where the segregating distance of the aggregate became shorter. This is mainly because, as soon as casting is over and before setting, gravity may induce sedimentation of the coarsest particles. The yield stress of a thixotropic cement paste will keep increasing once at rest. Its apparent viscosity will then increase and will be sufficient to prevent the particles from segregating.

## 5. Discussion

In this paper, a casting mold was constructed using COMSOL finite element simulation for small-scale as well as full-scale casting. The effects of the number and layout of the outlets and the speed of casting, as well as the surface tension, yield stress, and the plastic viscosity of the concrete, on the casting compactness were investigated. The effects of the yield stress and plastic viscosity of concrete on aggregate segregation were also investigated.

### 5.1. Impact of Casting Parameters

In the small-scale mold, the final debonding area decreased significantly as the number of air outlets increased. When the number of outlets was one or two, there was eventually an obvious debonding area, and when the number of outlets was three, no debonding area existed no matter how the outlets are arranged. Taking the three-outlets mold as an example, with the increase in the casting speed, the debonding area increased significantly. When the casting speed was less than 10 mm/s, there was no debonding area, mainly because a slower speed allowed the SCC to have enough time to flow uniformly. 

The influence of the number and layout of air outlets on the casting compactness in the full-scale mold was different from that in the small-scale mold. As the number of air outlets increased, the final debonding area decreased. However, in the full-scale mold, no matter if the number of outlets changed from one to three, there was still a debonding area in the final mold. This was mainly because in the full-scale mold, although it had the same layout of outlets as in the small-scale mold, the distance between every two outlets was still very large, so it was difficult to use the results of the small-scale mold as a guideline for the full-scale mold. Then, the effect of casting speed was studied, and it was found that reducing the speed still could not reduce the debonding area. On this basis, six small outlets were added around the large outlets, and this layout successfully eliminated the debonding area. 

Thus, in the small-scale mold, one can construct three air outlets and guarantee a low casting speed (less than 10 mm/s). However, in a full-scale mold, one may consider adding small outlets on the basis of three large outlets to increase the airflow (e.g., two outlets between each of the three large outlets). 

### 5.2. Impact of Material Properties

The surface tension of the concrete affects the air content of the concrete. When the surface tension was too low (less than 0.02 N/m), the SCC easily involved a large amount of air, so the final air content was higher. When the surface tension was too high, less air was entrained, but the air bubbles were difficult to burst and remained in the concrete. When the surface tension was more than 0.03 N/m, it could reduce the strength and stability of the bubble film, accelerate the exclusion of the liquid around the bubble, and make the liquid film thinner and rupture, thus eliminating the bubbles and decreasing the air content of the SCC. 

The yield stress and plastic viscosity of concrete will also affect the compactness of casting. When the yield stress and plastic viscosity were too low (25 Pa and 25 Pa·s), the SCC was too thin and could easily involve a large amount of air, resulting in a high air content. When the yield stress and plastic viscosity were too high (100 Pa and 100 Pa·s), the SCC was too viscous and it was difficult to flow uniformly on the surface, so the debonding area would eventually form. Only when the yield stress and plastic viscosity were moderate (50 Pa and 50 Pa·s) did the SCC flow easily on the surface and not trap large amounts of air.

From the above analysis of the debonding area, it can be seen that this debonding phenomenon was mainly due to the insufficient number of outlets or excessive material viscosity and yield stress, resulting in air not being removed from the mold in a timely manner, so that aggregation occurred at the top, which led to the formation of a debonding phenomenon.

Finally, the relationship between the aggregate segregation properties and the rheological properties of the interstitial cement paste was studied. Taking the largest aggregate particle size of 2 cm as an example, it was found that when the yield stress of the cement was higher than 3 Pa, the aggregate could be stable in the cement paste, and when the yield stress was lower than 3 Pa, the aggregate would segregate. When segregation occurred, as the yield stress of the paste kept increasing due to its thixotropic feature, the velocity of aggregate segregation was greatly reduced compared to the case where there was no thixotropic feature. Thus, it makes sense to use the thixotropic model to consider the segregation properties of aggregates in cement.

### 5.3. Practical Implications in Construction Scenarios

In the real world, the design of different concrete mixing proportions is inevitable in engineering, and they will affect the important rheological parameters in this paper, including yield stress and plastic viscosity. Concrete with too high a flowability (yield stress and plastic viscosity less than 50 Pa and 50 Pa·s) will not debond from the mold, but too low a yield stress will lead to segregation of the aggregates and is not suitable for practical engineering. Concrete with too low a flowability (yield stress and plastic viscosity great than 50 Pa and 50 Pa·s) will not easily remove air from the mold, resulting in debonding of the concrete from the mold, which is also unsuitable for practical engineering. However, concrete with a moderate flowability (yield stress and plastic viscosity are 50 Pa and 50 Pa·s) can keep the aggregates from segregating, and at the same time, the concrete will not debond from the mold. Therefore, a yield stress and plastic viscosity of 50 Pa and 50 Pa·s is an excellent choice for practical engineering.

In addition, considering the problems of bonding between fresh concrete and a steel shell, and the shrinkage of hardened concrete, some surfactants with a shrinkage-reduction function are usually added, which can reduce the surface tension of concrete significantly (as low as 0.03–0.063 N/m). According to the study of surface tension in this paper (see Section 3.3), a proper reduction in surface tension can help to improve the compactness of the casting.

However, this paper mainly considered the early rheological properties of concrete and its debonding from the steel shell. In future work, it is necessary to further consider the debonding of hardened concrete from the steel shell due to shrinkage or cracking. In addition, whether switching to a different casting mold, such as wood or plastic, will produce different results from steel shell casting under the same casting parameters needs further verification and simulation.

## 6. Conclusions

In this paper, a steel shell mold for the concrete casting of immersed tube tunnels was constructed using COMSOL finite element software (version 6.1). The influence of the number and layout of outlets and the speed of casting, as well as the surface tension, yield stress, and the plastic viscosity of SCC on the casting compactness and air content are investigated. In addition, the effects of the yield stress and plastic viscosity of the cement paste on aggregate segregation were also discussed. The main conclusions are as follows.

When the surface tension of SCC is greater than 0.03 N/m, there is excellent casting compactness. Furthermore, the addition of an appropriate amount of surfactant to SCC can help to reduce the air content of SCC, which can further improve the casting compactness. Moreover, when the yield stress and plastic viscosity are 50 Pa and 50 Pa·s, respectively, the SCC exhibits great casting compactness. 

Additionally, the layout of the outlets has different effects on casting compactness in the full-scale and small-scale molds. In small-scale mold, the compactness of casting is decent when the number of outlets is three and the casting speed is lower than 10 mm/s. However, it is necessary to incorporate three large and six small outlets paired with a casting speed of 3 cm/s in the full-scale mold for superior compactness. 

In addition, when the yield stress of cement paste exceeds 3 Pa, there is no segregation of aggregates. In the case of aggregate separation, the rate of aggregate segregation decreases as the viscosity of the cement paste increases. On this basis, when the thixotropic properties of the cement paste are taken into account, it is found that the rate of aggregate segregation decreases significantly.

In future work, further aspects of the geometry of casting molds will be investigated. In addition, we need to check the morphology of the aggregate and the particle size distribution on the casting compactness between SCC and the mold.

## Figures and Tables

**Figure 1 materials-16-06856-f001:**
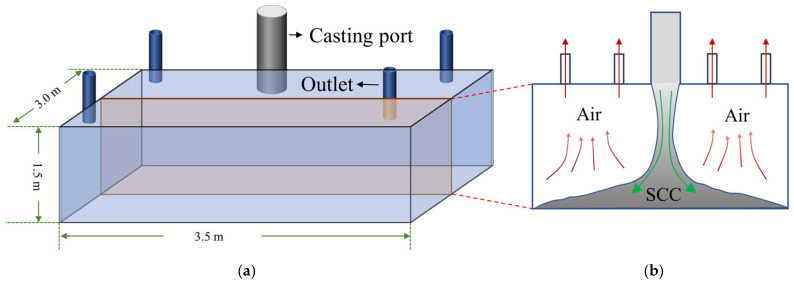
Schematic of the full-scale mold. (**a**) Three-dimensional mold (with length of 3.5 m, width of 3 m, and height of 1.5 m); (**b**) two-dimensional mold (The red arrow indicates the direction of air flow, and the green arrow indicates the direction of SCC flow).

**Figure 2 materials-16-06856-f002:**
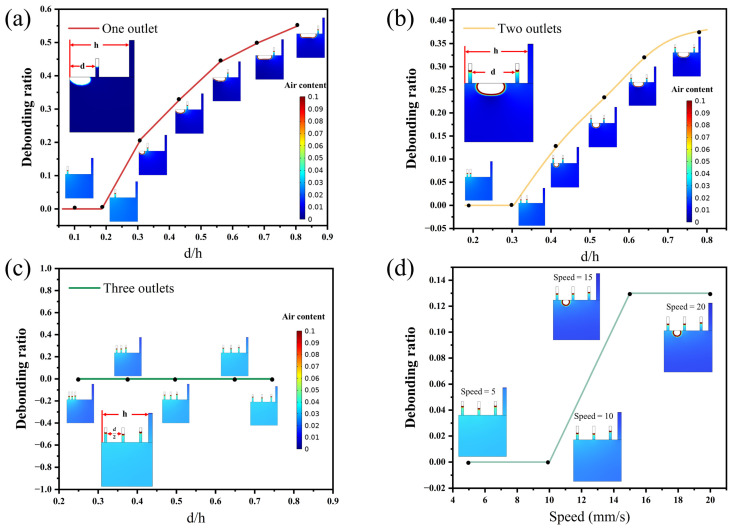
Influence of the number of outlets, layout, and casting speed on the casting compactness. (**a**) One outlet (casting speed of 5 mm/s), (**b**) two outlets (casting speed of 5 mm/s), and (**c**) three outlets (casting speed of 5 mm/s); (**d**) casting speed changes from 5 mm/s to 20 mm/s. Note: the size of the simulated mold corresponds to 35 cm × 15 cm in the real case.

**Figure 3 materials-16-06856-f003:**
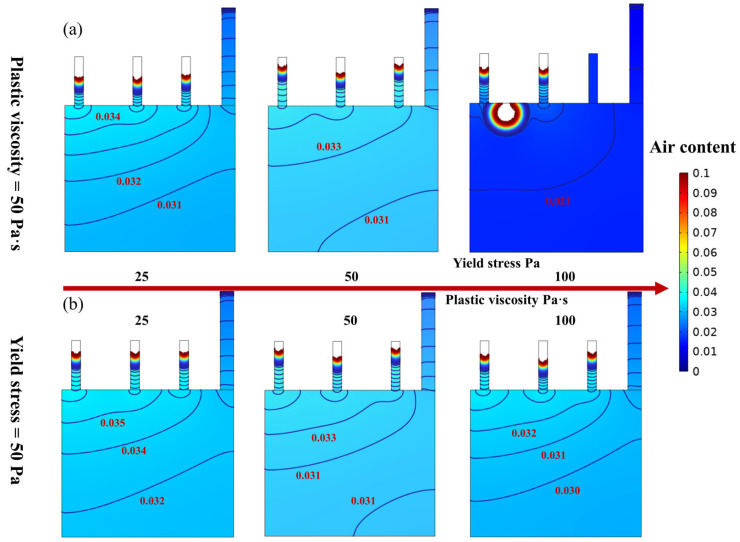
Effect of yield stress and plastic viscosity from 25 Pa, 25 Pa·s to 100 Pa 100 Pa·s on casting compactness and air content (the casting speed is 5 mm/s). Note: the size of the simulated mold corresponds to 35 cm × 15 cm in a real case. (**a**) Effect of yield stress of 25 Pa, 50 Pa and 100 Pa on casting compactness and air content for a plastic viscosity of 50 Pa·s for SCC; (**b**) Effect of plastic viscosity of 25 Pa·s, 50 Pa·s and 100 Pa·s on casting compactness and air content for a yield stress of 50 Pa for SCC.

**Figure 4 materials-16-06856-f004:**
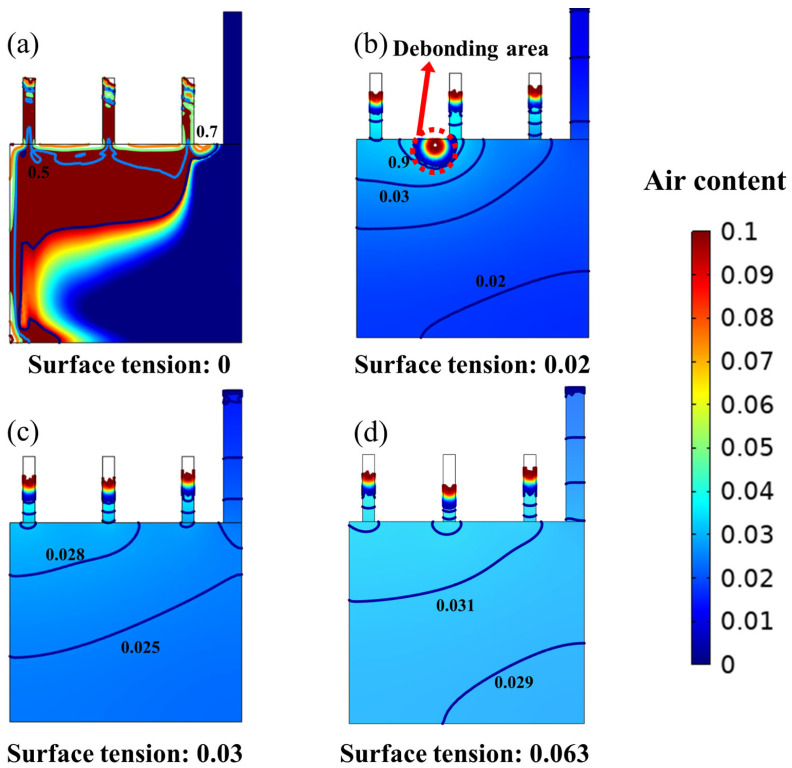
Effect of change in surface tension from 0 N/m to 0.063 N/m on the air content of SCC (casting speed is 5 mm/s). Note: the size of the simulated mold corresponds to 35 cm × 15 cm in the real case. (**a**) Surface tension is 0 N/m, (**b**) surface tension is 0.02 N/m, and (**c**) surface tension is 0.03 N/m; (**d**) surface tension is 0.063 N/m.

**Figure 5 materials-16-06856-f005:**
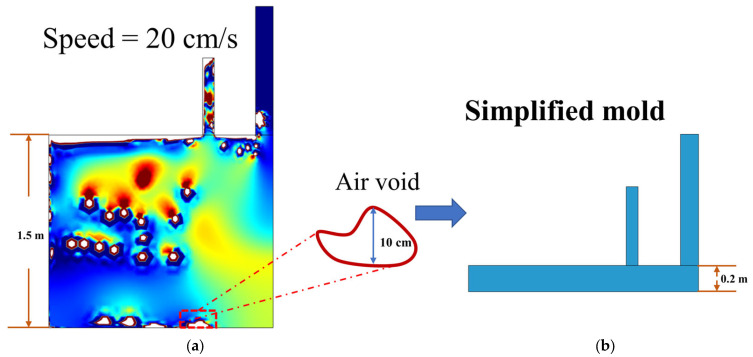
Simplification of the full-scale mold. (**a**) Casting results in the full-scale mold for a casting rate of 20 cm/s; (**b**) simplified full-scale mold considering the maximum air void size. Note: the size of the simulated mold corresponds to 3.5 m × 1.5 m in the real case.

**Figure 6 materials-16-06856-f006:**
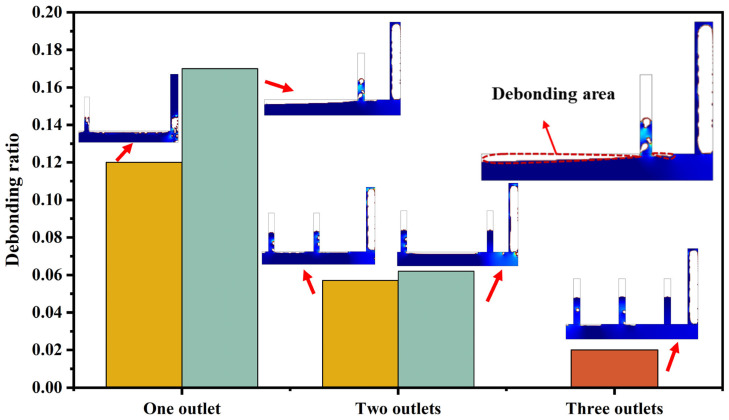
Influence of the number of outlets and their layout (one to three outlets) on casting compactness in the full-scale mold (casting rate maintained at 3 cm/s). Note: the size of the simulated mold corresponds to 3.5 m × 1.5 m in real case.

**Figure 7 materials-16-06856-f007:**
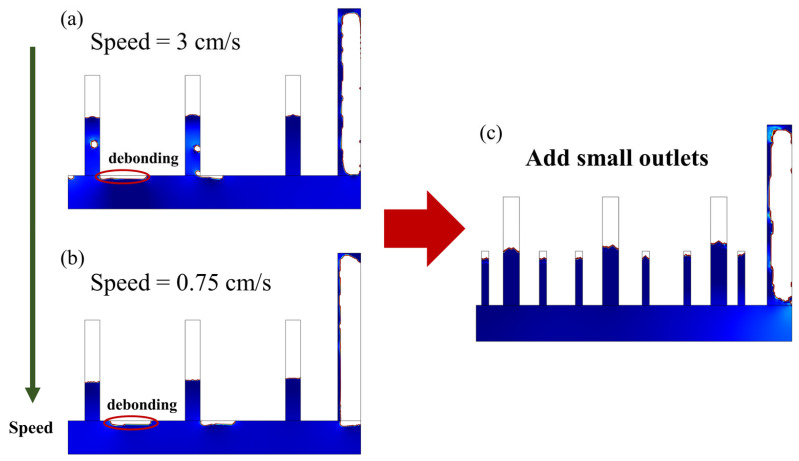
Improvement in outlet layout in the simplified full-scale mold (**a**) Casting results for casting speeds of 3 cm/s, and (**b**) casting results for casting speeds of 0.75 cm/s; (**c**) casting results with the additional six small outlets. Note: the size of the simulated mold corresponds to 3.5 m × 1.5 m in a real case.

**Figure 8 materials-16-06856-f008:**
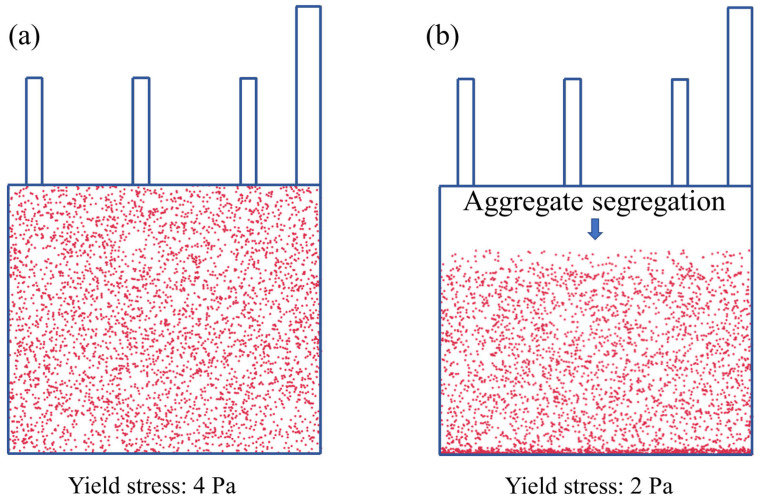
Effect of yield stress of 4 Pa and 2 Pa on the segregation of aggregates. Note: the size of the simulated mold corresponds to 3.5 m × 1.5 m in a real case. (**a**) Yield stress is 4 Pa; (**b**) yield stress is 2 Pa.

**Figure 9 materials-16-06856-f009:**
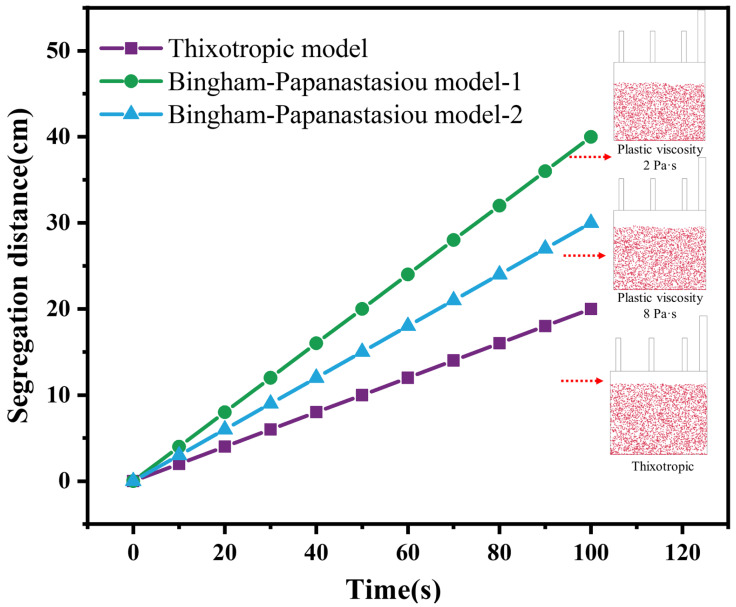
Effect of thixotropic property on aggregate segregating velocity for a yield stress of 2 Pa. Note: paste viscosities of 2 Pa·s and 8 Pa·s were used in the Bingham–Papanastasiou model (without considering the thixotropic feature); paste viscosity of 2 Pa·s was used in the thixotropic model (considering thixotropic property). The size of the simulated mold corresponds to 3.5 m × 1.5 m in the real case.

## Data Availability

Data sharing is not applicable for this article.

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
