# Peer review of "Casting Simulation of Large-Volume Fluid Cementitious Materials: Effect of Material Properties and Casting Parameters"

_materials, 2023, doi:10.3390/ma16216856_

Round 1

Reviewer 1 Report

Thank you for this contribution. This is an interesting and timely manuscript that covers casting simulations of cementitious materials. The conducted analysis is typically standard and falls within the expected work from such a publication and hence the work merits publication. As such, the authors are invited to properly address the following items:

1. Please fix the "Error! Reference source not found"

2. Please add some discussion on the debonding aspects observed in the simulations. How does this debonding occur? What are the physical mechanisms behind it?

3. The details on the FE models are quite superficial and light. Please ensure that your model is properly described to enable interested researchers to extend and replicate your work. Special attention should be paid to specifics such as element type (DOFs), convergence criteria, and performance metrics. Did the authors consider geometric imperfections?

4. How about the effect of residual stresses?

5. How would the results of the analysis differ if the model was made from timber or plastic as commonly used in the field?

Author Response

Review 1

Thank you for this contribution. This is an interesting and timely manuscript that covers casting simulations of cementitious materials. The conducted analysis is typically standard and falls within the expected work from such a publication and hence the work merits publication. As such, the authors are invited to properly address the following items:

Response:

We sincerely thank the reviewer’s kindly words and positive comments. We have tried our best to respond the reviewer’s suggestions or questions point-by-point in the following. The revisions are highlighted in red background to distinguish from the previous manuscript.

  1. Please fix the "Error! Reference source not found"

Response to 1:

Thanks for the reviewer’s helpful suggestion, all errors of “shown in Error! Reference source not found” have been fixed in the text.

  1. Please add some discussion on the debonding aspects observed in the simulations. How does this debonding occur? What are the physical mechanisms behind it?

Response to 2:

Thanks for the reviewer’s nice question and constructive advice. We have mainly discussed in the previous section the influence of the physical properties of concrete and the process of casting on the debonding performance of concrete and mold. However, the physical mechanism behind the debonding phenomenon is not explained in the paper. Therefore, the authors further explain the debonding phenomenon and the physical mechanism of its generation in the text.

This debonding phenomenon is mainly due to the insufficient number of outlets or excessive material viscosity and yield stress, resulting in air not being removed from the mold in a timely manner, so that aggregation occurs at the top, which leads to the formation of debonding phenomenon. Please find the revised content in line 413

  1. The details on the FE models are quite superficial and light. Please ensure that your model is properly described to enable interested researchers to extend and replicate your work. Special attention should be paid to specifics such as element type (DOFs), convergence criteria, and performance metrics. Did the authors consider geometric imperfections?

Response to 3:

Thanks for the reviewer’s careful reading and nice questions. And the authors have further explained the modeling process.

Since we are mainly concerned with debonding between concrete and steel shells during large volume casting, so concrete can be viewed as a homogeneous fluid. This casting process can be realized using the "Laminar two-phase flow" module of the COMSOL software, in which the phase field method is used to capture the air-water interface when solving the Navier–Stokes equations. In addition, the interface position is tracked by the phase field variable equation and the mixing energy density equation.

Furthermore, during the modeling process, the authors have taken into account the modeling to be free of geometric imperfections when modeling for practical engineering. The model is meshed with a free tetrahedral mesh with 33792 grid cells. A segregated solver is used to solve the velocity field, pressure field and phase field variations separately. For details, please refer to line 141 of the text.

  1. How about the effect of residual stresses?

Response to 4:

Thanks for the reviewer’s nice question and helpful suggestion. If the authors understand correctly, the residual stress referred to by the reviewer is the shrinkage stress of concrete after hardening, of course, it will affect the overall volume of the concrete after hardening, which will lead to the potential debonding of the concrete with the molds. To address the problem, in practice, the concrete mix ratio needs to be adjusted, such as the addition of shrinkage reducers, expanders and other additives, they will to a certain extent affect the early rheological properties of concrete, including yield stress, plastic viscosity and so on. These main parameters are analyzed in the paper, and this paper focuses on the early process of casting, for the late debonding phenomenon lies in the concrete mix ratio control.

  1. How would the results of the analysis differ if the model was made from timber or plastic as commonly used in the field?

Response to 5:

Thanks for the reviewer’s nice questions. The material chosen for this paper, to be precise, does not affect the simulation results, and such a large volume of concrete pouring once again uses steel molds rather than wood or plastic with poorer mechanical properties. Indeed, the material of the mold will affect the concrete contact with it to a certain extent, mainly due to the hydrophilicity of the mold. And the authors of this paper believe that the local contact of such a large volume of concrete pouring does not have much effect on the overall distribution of concrete in the mold. But in-depth research needs to be carried out in the future.

Reviewer 2 Report

Clarity and Structure:

The text lacks a clear introduction and conclusion. Consider adding introductory and concluding paragraphs to provide context and summarize the key points discussed.

Break down the text into smaller paragraphs for better readability. Each paragraph should focus on a specific aspect or point to enhance clarity.

Explanation of Technical Terms:

Some technical terms and concepts related to tunnel construction and concrete casting might be complex for readers unfamiliar with the field. Consider providing brief explanations or using simpler language where possible to enhance accessibility.

Integration of Citations:

The citation format seems to be incomplete, and the references are not integrated properly into the text. Ensure that the citations are appropriately linked to the relevant information in the text.

The text describes the outcomes of various experiments but lacks a cohesive analysis of the results. Discuss the implications of the findings in the context of practical applications. For example, explain how the optimal values of yield stress and plastic viscosity (50 Pa·s and 50 Pa) are determined and why these values are suitable for casting.

Include more technical details about the experimental setup, the methodology used, and the specific observations made during the experiments. Providing detailed information will enhance the scientific rigor of the study.

Everywhere, there is an Error! Reference source not found. Kindly check the error.

Discuss the practical implications of the findings in real-world construction scenarios. How might the optimized casting parameters influence actual tunnel construction projects? Addressing this aspect would enhance the relevance of the study's conclusions for industry professionals and researchers.

Minor editing of English language required

Author Response

Reviewer 2:

Response:

The authors are sincerely grateful for the reviewer’s careful reading and constructive comments on our manuscript, and we have made our best effort to improve the paper. A point-by-point detailed response to the reviewer’s remarks has been prepared and reported hereinafter. The revisions are highlighted in red background to distinguish from the previous manuscript.

  1. Clarity and Structure:

1.1The text lacks a clear introduction and conclusion. Consider adding introductory and concluding paragraphs to provide context and summarize the key points discussed.

Response to 1.1:

Thanks for the reviewer’s careful reading and nice questions. In this article, we focus on the problem of debonding between concrete and steel shells during concrete placement in immersed tube tunnels. Simulations were used to investigate the effects of the physical properties of fresh concrete and the outlets of the casting mold on the casting compactness and bonding. Considering that many readers are not familiar with the construction of immersed tunnel, we have further added an introduction to the construction of immersed concrete and simulation setup in the introduction section. For details, please refer to lines 33, 41, and 106.

In addition, in the discussion chapter, we further explain the physical phenomenon of debonding that we have observed. As well as further discussing the feasibility of simulations results in practical engineering. Please refer to lines 413 and 428 for more details.

1.2 Break down the text into smaller paragraphs for better readability. Each paragraph should focus on a specific aspect or point to enhance clarity.

Response to 1.2:

Thanks for the reviewer’s helpful suggestion, some of the long paragraphs have been divided into subparagraphs. Please refer to line 147,218 and 355

  1. Explanation of Technical Terms:

Some technical terms and concepts related to tunnel construction and concrete casting might be complex for readers unfamiliar with the field. Consider providing brief explanations or using simpler language where possible to enhance accessibility.

Response to 2:

Thanks for the reviewer’s careful reading and nice questions. It is true that many people are not very familiar with the construction of immersed concrete tunnels, so we have added some explanations of common terms used in the construction of immersed tunnels. The main terms include "immersed tube method" and "debonding area". Please refer to lines 41, 72 and 204 for details.

  1. Integration of Citations:

3.1 The citation format seems to be incomplete, and the references are not integrated properly into the text. Ensure that the citations are appropriately linked to the relevant information in the text.

Response to 3.1:

Thanks for the reviewer’s high-quality comments and nice question. After we checked the references, there were indeed some references that were not very relevant to the text content. Therefore, we replaced and added some references that were more relevant to the text. Please refer to [1][2][6][29].

3.2 The text describes the outcomes of various experiments but lacks a cohesive analysis of the results. Discuss the implications of the findings in the context of practical applications. For example, explain how the optimal values of yield stress and plastic viscosity (50 Pa·s and 50 Pa) are determined and why these values are suitable for casting.

Response to 3.2:

Thanks for the reviewer’s careful reading and nice question. In our text, we have directly chosen the plastic viscosity and yield stress as 50 Pa·s and 50 Pa without giving an explanation. This is indeed a lack of rigor. Therefore, in our paper, we have added the sources of the yield stress and plastic viscosity for the selection of 50 Pa and 50 Pa·s. The main reference is Wallevik's study, combined with the recommended yield stress and plastic viscosity plots drawn by Wallevik. Our choice of plastic viscosity and yield stress of 50 Pa·s and 50 Pa is suitable for practical engineering. Please refer to line 152 in the text for details

3.3 Include more technical details about the experimental setup, the methodology used, and the specific observations made during the experiments. Providing detailed information will enhance the scientific rigor of the study.

Response to 3.3:

Thanks for the reviewer’s careful reading and nice question. Based on the reviewers' comments, the specific methods used in the simulation and the details of the simulation setup are explained in more detail, mainly including the principles on which the physical fields we used, as well as the model meshing, solver selection, etc. In the hope of providing the reader with a clear guideline and facilitating validation of the results we have concluded. Please refer to line 106 and 141 for details.

In addition, we further discuss the physical mechanisms behind the phenomenon of concrete and mold debonding: debonding phenomenon is mainly due to the insufficient number of outlets or excessive material viscosity and yield stress, resulting in air not being removed from the mold in a timely manner, so that aggregation occurs at the top, which leads to the formation of debonding phenomenon. Please refer to line 413 for details.

  1. Everywhere, there is an Error! Reference source not found. Kindly check the error. 

Response to 4:

Thanks for the reviewer’s careful reading, all errors of “shown in Error! Reference source not found” have been fixed in the text.

  1. Discuss the practical implications of the findings in real-world construction scenarios. How might the optimized casting parameters influence actual tunnel construction projects? Addressing this aspect would enhance the relevance of the study's conclusions for industry professionals and researchers.

Response to 5:

Thanks for the reviewer’s high-quality comments and nice question. Indeed, it is of great importance to integrate the results of finite element simulations with reality. Therefore, in the Discussion section, the authors of this paper further discuss how simulation results can guide practical engineering. An appropriate reduction in surface tension (surface tension between 0.03 and 0.063 N/m) reduces the debonding of the concrete from the mold. In addition, a moderate yield stress and plastic viscosity (yield stress and plastic viscosity of 50 Pa, 50 Pa·s) reduces the segregation properties of the concrete and reduces the debonding of the concrete from the mold. Please refer to line 428 for details

Reviewer 3 Report

The Article adequately presents the state of the art, methodology, results, and discussion. I have some suggestions:

1. A review of the text is necessary as there are several errors in inserting Figures in the text: "shown in Error! Reference source not found".

2. The figures are disproportionate to the text: the words must be at most the same size as those in the text.

3. Item 5 “Summary”: the name is inappropriate; the content should be incorporated into the discussions of the results whether it is necessary 

4. According to the authors “In this paper, a casting mold is constructed using COMSOL finite element simulation for small-scale as well as full-scale casting.” The justification for the study is: “.....Although these software programs have faster operation rates for single-field calculations, the concrete is a complex mixture, requiring considerations like the two-phase flow between concrete and air and the relative motion between the interstitial cement paste and aggregates in concrete mixtures. Therefore, multi-physics field simulation is involved. COMSOL software has a significant advantage in multi-physics field simulation, accurately handling couplings between multiple physical fields and offering a more comprehensive custom interface for fluid parameters and grid division.” The text must clearly inform how to validate the results of the new simulation and whether it is more appropriate.

Author Response

Review 3

The Article adequately presents the state of the art, methodology, results, and discussion. I have some suggestions:

Response:

We sincerely thank the reviewer’s kindly words and positive comments. We have tried our best to respond the reviewer’s suggestions or questions point-by-point in the following. The revisions are highlighted in red background to distinguish from the previous manuscript.

  1. A review of the text is necessary as there are several errors in inserting Figures in the text: "shown in Error! Reference source not found".

Response to 1:

Thanks for the reviewer’s helpful suggestion, all errors of “shown in Error! Reference source not found” have been fixed in the text.

  1. The figures are disproportionate to the text: the words must be at most the same size as those in the text.

Response to 2:

Thanks for the reviewer’s nice question and constructive advice, the size proportions of figures and text have been changed throughout the paper.

  1. Item 5 “Summary”: the name is inappropriate; the content should be incorporated into the discussions of the results whether it is necessary 

Response to 3:

Thanks for the reviewer’s helpful suggestion. After careful consideration, the authors do agree with the reviewers' comments. The title "Summary" has been changed to "Discussion". For details, please refer to line 370 of the article.

  1. According to the authors “In this paper, a casting mold is constructed using COMSOL finite element simulation for small-scale as well as full-scale casting.” The justification for the study is: “.....Although these software programs have faster operation rates for single-field calculations, the concrete is a complex mixture, requiring considerations like the two-phase flow between concrete and air and the relative motion between the interstitial cement paste and aggregates in concrete mixtures. Therefore, multi-physics field simulation is involved. COMSOL software has a significant advantage in multi-physics field simulation, accurately handling couplings between multiple physical fields and offering a more comprehensive custom interface for fluid parameters and grid division.” The text must clearly inform how to validate the results of the new simulation and whether it is more appropriate.

Response to 4:

Thanks for the reviewer’s careful reading and nice questions. Based on the reviewers' comments. In the main text, we further explain the modeling process, and the principles on which the physical fields we used, as well as the model meshing, solver selection, etc. In the hope of providing the reader with a clear guideline and facilitating validation of the results we have concluded. The main additions are as follows:

Since we are mainly concerned with debonding between concrete and steel shells during large volume casting, so concrete can be viewed as a homogeneous fluid. This casting process can be realized using the "Laminar two-phase flow" module of the COMSOL software, in which the phase field method is used to capture the air-water interface when solving the Navier–Stokes equations. In addition, the interface position is tracked by the phase field variable equation and the mixing energy density equation.

In addition, during the modeling process. The model is meshed with a free tetrahedral mesh with 33792 grid cells. A segregated solver is used to solve the velocity field, pressure field and phase field variations separately.

Please, see lines 106 and 142 of the text for details of the additions.

Round 2

Reviewer 1 Report

.